# A comprehensive molecular characterization of the 8q22.2 region reveals the prognostic relevance of OSR2 mRNA in muscle invasive bladder cancer

Daniel Uysal[1], Karl-Friedrich Kowalewski[1], Maximilian Christian Kriegmair[1], Ralph Wirtz[2], Zoran V. Popovic[3], Philipp Erben[1]*

1 Department of Urology and Urosurgery, Medical Faculty Mannheim, University of Heidelberg, Mannheim, Germany, 2 STRATIFYER Molecular Pathology GmbH, Köln, Germany, 3 Institute of Pathology, Medical Faculty Mannheim, University of Heidelberg, Mannheim, Germany

* philipp.erben@medma-uni.heidelberg.de

**Data Availability Statement:** All relevant data are within the manuscript and its Supporting Information files. In silico data have been accessed

## Abstract

Technological advances in molecular profiling have enabled the comprehensive identification of common regions of gene amplification on chromosomes (amplicons) in muscle invasive bladder cancer (MIBC). One such region is 8q22.2, which is largely unexplored in MIBC and could harbor genes with potential for outcome prediction or targeted therapy. To investigate the prognostic role of 8q22.2 and to compare different amplicon definitions, an in-silico analysis of 357 patients from The Cancer Genome Atlas, who underwent radical cystectomy for MIBC, was performed. Amplicons were generated using the GISTIC2.0 algorithm for copy number alterations (DNA_Amplicon) and z-score normalization for mRNA gene over-expression (RNA_Amplicon). Kaplan-Meier survival analysis, univariable, and multivariable Cox proportional hazard ratios were used to relate amplicons, genes, and clinical parameters to overall (OS) and disease-free survival (DFS). Analyses of the biological functions of 8q22.2 genes and genomic events in MIBC were performed to identify potential targets. Genes with prognostic significance from the in silico analysis were validated using RT-qPCR of MIBC tumor samples (n = 46). High 8q22.2 mRNA expression (RNA-AMP) was associated with lymph node metastases. Furthermore, 8q22.2 DNA and RNA amplified patients were more likely to show a luminal subtype (DNA_Amplicon_core: p = 0.029; RNA_Amplicon_core: p = 0.01). Overexpression of the 8q22.2 gene OSR2 predicted shortened DFS in univariable (HR [CI] 1.97 [1.2; 3.22]; p = 0.01) and multivariable in silico analysis (HR [CI] 1.91 [1.15; 3.16]; p = 0.01) and decreased OS (HR [CI] 6.25 [1.37; 28.38]; p = 0.0177) in RT-qPCR data analysis. Alterations in different levels of the 8q22.2 region are associated with manifestation of different clinical characteristics in MIBC. An in-depth comprehensive molecular characterization of genomic regions involved in cancer should include multiple genetic levels, such as DNA copy number alterations and mRNA gene expression, and could lead to a better molecular understanding. In this study, OSR2 is identified as a potential biomarker for survival prognosis.

through cBioPortal (URL: https://www.cbioportal.org/results/oncoprint?genetic_profile_ids_PROFILE_COPY_NUMBER_ALTERATION=blca_tcga_pub_2017_gistic&genetic_profile_ids_PROFILE_MRNA_EXPRESSION=blca_tcga_pub_2017_rna_seq_v2_mrna_median_Zscores&cancer_study_list=blca_tcga_pub_2017&Z_SCORE_THRESHOLD=2.0&RPPA_SCORE_THRESHOLD=2.0&data_priority=0&profileFilter=0&case_set_id=blca_tcga_pub_2017_3way_complete&gene_list=RNF19A%250ASPAG1%250ARGS22%250APOLR2K%250AFBXO43%250ACOX6C%250AVPS13B%250ASTK3%250AOSR2%250AKCNS2%250ARPL30%250ARIDA%250APOP1%250ANIPAL2%250AERICH5%250ASNORA72&geneset_list=%20&tab_index=tab_visualize&Action=Submit) and the XENA browser (URL:https://xenabrowser.net) and are deposited as a supplementary file named S1 Document. RT-qPCR data from our own Lab are deposited as a supplementary file named S2 Document.

**Funding:** R. M. Wirtz is the CEO and employee of STRATIFYER Molecular Pathology GmbH. The XTRAKT FFPE Kit for RNA Isolation was developed by STRATIFYER and is commercially available on the German market. This does not alter our adherence to PLOS ONE policies on sharing data and materials. STRATIFYER Molecular Pathology GmbH did not play a role in the funding of this study, nor in the study design, data collection and analysis, decision to publish, or preparation of the manuscript and only provided financial support in the form of R.M. Wirtz salary.

**Competing interests:** R. M. Wirtz is the CEO and employee of STRATIFYER Molecular Pathology GmbH. The XTRAKT FFPE Kit for RNA Isolation was developed by STRATIFYER and is commercially available on the German market. This does not alter our adherence to PLOS ONE policies on sharing data and materials.

## Introduction

Bladder cancer is one of the most common urologic malignancies with estimated 80,470 new cases and 17,670 new deaths in the United States in 2019 [1]. Despite treatment advances, muscle invasive bladder cancer (MIBC) is associated with a poor 5-year survival rate of 40–60% [2]. To develop targeted therapies for MIBC, much effort has been directed at understanding the pathogenesis of this disease. Technological advances in molecular profiling, such as high-throughput sequencing technologies, have augmented cytogenetic, fluorescence in situ hybridization (FISH) and array-comparative-genomic-hybridization (aCGH) studies. These results have created an additional complexity through the generation of an unprecedented series of copy number alterations (CNA) and mRNA gene expression data, in addition to classical oncogene mutations [3–7]. The complexity of this molecularly altered landscape and its potential interactions requires an in-depth comprehensive analysis and correlation with clinical phenotype and prognosis [8].

Common regions of gene amplification on chromosomes are known for various solid cancers and hematologic malignancies [9]. In an effort to define these regions of co-amplified neighboring genes, many authors have used the term 'amplicon' [10], but the exact definition of an amplicon is currently a topic of discussion. Bignell et al. defined an amplicon as "a somatically acquired increase in copy number of a restricted genomic region," which "is often found in cancer cells as a mechanism of increasing the transcript and therefore, protein levels of dominantly acting cancer genes"[11–13]. Based on this definition, the GISTIC2.0 algorithm has been used to select potential target genes in a specific region with a CNA of $\geq 2$. If co-amplification of these genes is detected, this region is referred to as an amplicon [14, 15]. Since amplicons are usually investigated in the context of survival or prognostic influence on cancer, using CNA may underestimate functionally relevant levels of the genetic code, such as mRNA and protein expression. As current methods to measure protein expression are limited, mRNA expression is suggested as the closest and most accurate surrogate for gene activity [16].

The mRNA-based approach for selecting amplicons has been applied by Luo et al., who used a multigene mRNA expression signature to identify *FGFR1*-amplified estrogen receptor-positive (ERfl) breast tumors [17]. While this study and other research have mainly focused on amplicons in breast and gastric cancer, limited knowledge exists for bladder cancer [18–20].

Hurst et al. investigated genome-wide copy number alterations in urothelial carcinoma and found a 30.4% gain frequency for the 8q22.2-q22.3 region [21]. High-level amplification in this region has been associated with more aggressive types of urothelial carcinoma [21–24]. However, current knowledge of 8q22.2's oncogenic properties in MIBC is limited [21–24].

i.  The primary aim of this study is to provide a method for a comprehensive molecular characterization of the 8q22.2 region in bladder cancer using in silico CNA and mRNA gene expression data.

ii.  A second aim of this study is to compare the CNA and mRNA amplicon structure of 8q22.2, investigate the prognostic value of 8q22.2 amplification on survival, and identify predictive genes. Prognostically significant genes are validated using RT-qPCR in an independent cohort of MIBC tumor samples.

iii.  Finally, this study is also designed to explore whether the number of chromosomal alterations is associated with a distinct group of patients with clinically or histologically defined parameters.

## Material and methods

### TCGA cohort

Data for the first muscle invasive bladder cancer (MIBC) cohort for this study was derived from The Cancer Genome Atlas and has been produced in earlier analyses [7]. Clinicopathological data on the cohort, CNA data, mRNA data, and mutation data were downloaded from the open access portal, cBioPortal (https://www.cbioportal.org) provided by the Memorial Sloan Kettering Cancer Center [25, 26].

Molecular subtypes and focal amplifications and deletions were downloaded from a supplementary excel file from the TCGA analysis performed by Robertson et al., named S1 Table: Clinical and Molecular covariates, Related to STAR Methods [7]. Focal gains and amplifications, as well as focal losses and deletions, were grouped together and analyzed as events of genomic amplification or deletion.

CNA data were generated using Affymetrix SNP6.0 arrays [7]. 'Broad automated' mutation data files, generated according to the MutSigCV algorithm, were obtained from the XENA browser (https://xenabrowser.net/) [27, 28].

Patients undergoing neoadjuvant chemotherapy and with an unknown tumor stage (Tx) or T<2 were excluded from further analysis, yielding a final study population of 357 patients (73% male, 27% female, median age 69, T3/4 68%, N+ 36%) (Fig 1, S1 Document).

Receipt of neoadjuvant chemotherapy and unknown tumor stage (Tx) or T<2 were defined as exclusion criteria. After exclusion, 357 patients remained for final analysis.

**Validation cohort.**   In silico findings were retrospectively validated in an independent MIBC ($\geq$2) cohort from the Clinic for Urology and Urosurgery at the University Hospital Mannheim. Formalin-fixed, paraffin-embedded (FFPE) tissue samples were obtained from 46 patients (24% female, with a median age of 72.5 years; 78% T3 and T4 tumors and 27% lymph node positive patients (N+)) (S2 Document). Histopathologic information has been provided by ZVP. Specimens were graded according to the most recent TNM classification (2017) and the WHO 2010 classification of genitourinary tumors. Prior to commencement the review board 2 of the University of Heidelberg approved the study under the number 2015-549N-MA, in accordance with the Declaration of Helsinki. All patients provided written informed consent for this study.

### TCGA bladder cancer copy number alterations and gene expression data

Varying degrees of co-amplified genes in the 8q22.2 region for patients in the TCGA bladder cancer cohort were visualized using a heatmap (oncoprint) (S1 Fig). For this study we only addressed high CN gains or amplifications and did not differentiate between balanced and unbalanced gains. We chose to define genes as amplified on the DNA level if they showed a GISTIC2.0 algorithm value $\geq$2, as reported by Mermel et al. [29].

Amplification at the mRNA level was defined as mRNA overexpression with a z-score-normalization value $\geq$2 (https://docs.cbioportal.org/5.1-data-loading/data-loading/file-formats/z-score-normalization-script). This value was based on the fact that a z-score of 2 (2 standard deviations from the mean) is the default setting used when downloading mRNA data from cBioPortal [26]. We further performed analyses with more or less restrictive z-scores (S1 and S2 Tables).

As in the work of Heidenblad et al., the size of the 8q22.2 amplicon was defined by the minimal common region (i.e., number of genes) of amplification present in a given percentage of the patient population [22]. Based on the presence of a regional amplification in 8q22.2, patients were divided into amplified (AMP) and non-amplified (NONAMP) groups.

We then compared the mRNA gene expression between AMP and NONAMP to determine if amplification of genes resulted in mRNA overexpression (S3 Table).

**Prognostic value of 8q22.2 and clinical parameters.** To identify whether the number of chromosomal alterations in a patient was associated with specific clinical criteria, clinicopathologic data for patients were compared for each amplicon definition. Respective parameters included age ($\geq$70 vs. <70), gender (male vs. female), tumor stage (T2 vs. T3/4), lymph node status (N0 vs. N+), and molecular subtype (luminal vs. basal). Molecular subtype analysis was further extended to include five mRNA expression based molecular subtypes, as described by Robertson et al. [7] (S4 Table). Univariable Cox proportional hazard ratios and Kaplan-Meier curves were used to assess overall survival (OS) and disease-free survival (DFS). Clinical variables and 8q22.2 amplicon definitions or individual genes (RNA_AMP_Gene) with a p-value of <0.2 were further subjected to a multivariable Cox regression analysis [30].

**Biological analysis and identification of target genes on 8q22.2.** Biological functions and molecular processes of genes of the 8q22.2 region were identified using UniProtKB (https://www.uniprot.org) and GeneCards (https://www.genecards.org) [31, 32] (S5 Table).

We further assessed, whether amplification at the 8q22.2 locus was associated with genomic events such as mutations and chromosomal aberrations, including amplifications and deletions.

A list of 18 highly mutated (i.e., mutations in more than ten percent of patients in the TCGA cell 2017 cohort) known oncogenes and tumor suppressor genes in MIBC, as reported by Robertson et al., were chosen for comparison between AMP and NAMP groups [7] (S6 and S7 Tables).

The frequency of amplifications and deletions of frequently altered genomic regions in MIBC were compared for the AMP and NAMP groups (S8 Table).

**Validation of prognostic markers with RT-qPCR.** Gene expression of *COX6C* and *OSR2* was quantified using reverse transcription quantitative real-time polymerase chain reaction (RT-qPCR) methodology, as described previously [33, 34]. Briefly, RNA was extracted from 10 μm thick FFPE tissue slides with the bead-based XTRAKT FFPE kit (Xtract® kit; STRA-TIFYER Molecular Pathology GmbH, Cologne, Germany). After lysing and purifying, nucleic acid isolates were treated with DNAse I. After DNA digestion RNA was transcribed with Super Script III reverse transcriptase (Thermo Fisher Scientific, Waltham, MA, USA) and sequence specific primers (S9 Table). Relative quantification of mRNA expression of *COX6C* (ENSG00000164919), *OSR2* (ENSG00000164920) and the reference gene *CALM2* (ENSG00000143933) was measured on a StepOnePlus Real-Time PCR system (Life Technologies, Karlsbad, California, USA) with the SuperScript III Platinum One-Step quantitative RT-PCR system (Invitrogen, Karlsruhe, Germany) [35]. To normalize Ct values the Ct Value of the housekeeping gene *CALM2* was subtracted from the Ct values of the target genes (ΔCt). mRNA expression was further reported as 40-ΔCt. [35–37]. Median, 1st and 3rd quartile 40-ΔCt values of *COX6C* and *OSR2* were used as cut-off points for survival analysis.

A protocol of the workflow used in this study can be accessed at protocols.io (http://dx.doi.org/10.17504/protocols.io.bmewk3fe).

## Statistics

All *p*-values were calculated for two sided tests. Spearman coefficient analysis was performed to assess gene correlations using a cutoff of $\geq$0.5 for positive correlations, including all statistically significant negative correlations (S10 Table). Contingency analysis was visualized using mosaic plots and differences between DNA_Amplicon_Core and RNA_Amplicon_Core definitions were compared using Fisher's exact test. Chi-Square tests were used to compare clinicopathologic data of patients for different amplicon definitions. Comparisons included age, gender, tumor stage, lymph node status, and molecular subtype.

Association with mutations and genomic events was investigated using Chi-Square tests, and the Bonferroni correction was used to correct for multiple testing.

Survival analyses were conducted using univariable Kaplan-Meier regressions and tested for significance with the log-rank and Wilcoxon tests. Multivariable analyses were performed using Cox-proportional hazard regression models. For results from the univariable analysis a p value cut-off of <0.2 was chosen to include relevant clinical or pathologic parameters that would have been missed with a more restrictive p value of <0.05 [30]. Variables for the multivariable analysis included significant (p<0.2) clinicopathological characteristics on univariable analysis (pT-Stage, pN-Stage, age, gender,) and genes (RNA_AMP_COX6C, RNA_AMP_OSR2), or amplicon definitions (DNA_Amplicon_Core, RNA_Amplicon_Core). Statistical analyses of numeric continuous variables were performed with non-parametric tests (Wilcoxon rank-sum test, Kruskal-Wallis test). Contingency analyses of nominal variables were performed with Pearson's chi-squared test. All statistical analyses were performed using GraphPad Prism 8.0 (GraphPad Software Inc., La Jolla, California, USA) and JMP SAS 14.0 (SAS, Cary, North Carolina, USA). Results were considered to be significant if significance levels lower than 0.05 were obtained (the exception being significance levels lower than 0.2 on univariable analysis that were included in the multivariable analysis).

## Results

### 8q22.2 is a highly amplified region in TCGA cohort and can be divided into a core and extended region

More than ten percent of patients in the Cancer Genome Atlas (TCGA) cohort show CNA amplifications within the 8q22.2 region (https://www.cbioportal.org). The 8q22.2 amplicon can further be defined by a minimal common or core region consisting of two genes, *RNF19A* (ENSG00000034677) and *SPAG1* (ENSG00000104450), that are present in all patients with amplifications in this region. This region can be extended to seven genes from *RNF19A* to *VPS13B* (ENSG00000132549), Ext1, and 15 genes from *RNF19A* to *ERICH5* (ENSG00000 177459), Ext2 (Table 1).

When using mRNA gene expression data only the core amplicon definition could be replicated due to a high variance in gene co-expression at the mRNA level.

### Histopathological and clinical analysis of 8q22.2

Patient data are described in Table 2, with patients having a median age of 69 years (IQR: 60;77). Most patients were male (male: 259, female: 98) with advanced tumor stages (T3/4 = 242, T2 = 115) and lymph node negative disease (N+ = 120, N0 = 214). Association of clinicopathological data showed that patients with high mRNA expression in the RNA_Amplicon_Core are more likely to harbor lymph node positive disease, as compared to RNA_NONAMP patients (p = 0.04) (Table 2). Additionally, the luminal subtype was identified more often in patients with AMP_Core, as compared to NONAMP_Core patients (DNA_AMP_Core: p = 0.029; RNA_AMP_Core: p = 0.01; DNA_Amplicon_Ext1: p = 0.048; DNA_Amplicon_Ext2: p = 0.04) (Table 2). With regard to age, gender, tumor stage, and adjuvant chemotherapy, no associations were detected (Table 2).

### Univariable and multivariable survival analysis (DFS and OS) of the 8q22.2 region and genes in the TCGA cohort

Kaplan-Meier analysis of 5-year DFS and 5-year OS could not detect a statistically significant prognostic influence of 8q22.2 amplicons, either at the DNA or RNA level on survival (S3 and

**Table 1. Core and extended 8q22.2 amplicons.**

| | Genes in region | Amplified patients (n = number) | % of AMP patients in relation to the cohort |
|---|---|---|---|
| DNA_Amplicon_Core | RNF19A and SPAG1 | n = 58 | 16% |
| RNA_Amplicon_Core | RNF19A and SPAG1 | n = 30 | 8% |
| DNA_Amplicon_Ext1 | RNF19A – VPS13B(seven genes) | n = 50 | 14% |
| DNA_Amplicon_Ext2 | RNF19A – ERICH5 (15 genes) | n = 37 | 10% |

Comparison of amplicons according to the extended amplified region and gene level. The percentage of AMP patients appears to be higher in DNA_Amplicons as compared to RNA_Amplicons.

S4 Figs). Univariable analyses of OS and DFS (HR (hazard ratio) [CI (95% confidence interval)]; p-value) are summarized in Table 3, with age ($\geq$70 vs. <70 (HR [CI] 1.63 [1.19; 2.23]; p = 0.002), T stage (T3/4 vs. T$\leq$2) (HR [CI] 1.99 [1.36; 2.92]; p<0.001), and lymph node status (N+ vs. N0) (HR [CI] 2.11 [1.52; 2.93], p<0.001) significantly influencing OS. T stage (HR [CI 2.59 [1.67; 4.03]; p<0.001) and lymph node status also (HR [CI] 2.59 [1.67; 4.03]; p<0.001) significantly influenced DFS.

A univariable analysis of individual genes showed that RNA_HIGH_COX6C was significantly associated with improved OS (HR [CI] 0.66 [0.43; 1]; p = 0.04), while RNA_HIGH_OSR2 was significantly associated with lower levels of DFS (HR [CI] 1.97 [1.2; 3.22]; p = 0.01) (Table 3, Figs 2 and 3). No significant associations could be detected for any of the other amplicon genes at the RNA level.

Upon multivariable analysis of OS, age (HR [CI] 1.76 [1.24; 2.49]; p = 0.002) and lymph node status (HR [CI] 2.05 [1.44; 2.91]; p<0.001) remained significant, while no significant influence could be shown for any of the other variables (Table 3).

T stage (HR [CI] 2.31 [1.41; 3.79]; p<0.001) and lymph node status (HR [CI] 2.01 [1.36; 2.92]; p<0.001) along with RNA_AMP_OSR2 (HR [CI] 1.91 [1.15; 3.16]; p = 0.01), were significant prognostic factors for the multivariable analysis of DFS. None of the other clinical and histopathologic variables significantly influenced DFS in both the univariable and multivariable models.

**Table 2. Patient demographics of AMP and NONAMP patients according to DNA_Amplicon and RNA_Amplicon definitions of amplicons in the 8q22.2 region.**

| Clinicopathologic variables | | | | | DNA_Amplicon_Core | | | RNA_Amplicon_ Core | | | DNA_Amplicon_Ext1 | | |
|---|---|---|---|---|---|---|---|---|---|---|---|---|---|
| | | DNA_Amplicon_Ext2 | | | | | | | | | | | |
| | | AMP | NONAMP | P-value | AMP | NONAMP | P-value | AMP | NONAMP | P-value | AMP | NONAMP | P-value |
| Age | $\geq$ 70 | 30 | 143 | 0.59 | 14 | 159 | 0.84 | 28 | 145 | 0.25 | 23 | 150 | 0.08 |
| | < 70 | 28 | 156 | | 16 | 168 | | 22 | 162 | | 14 | 170 | |
| Sex | male | 41 | 218 | 0.73 | 22 | 237 | 0.92 | 36 | 223 | 0.93 | 25 | 234 | 0.47 |
| | female | 17 | 81 | | 8 | 90 | | 14 | 84 | | 12 | 86 | |
| T stage | $\leq$2 | 17 | 98 | 0.61 | 13 | 102 | 0.17 | 15 | 100 | 0.72 | 11 | 104 | 0.73 |
| | 3/4 | 41 | 201 | | 17 | 225 | | 35 | 207 | | 26 | 216 | |
| N stage | N0 | 32 | 182 | 0.12 | 13 | 201 | **0.04** | 28 | 186 | 0.2 | 19 | 195 | 0.09 |
| | N+ | 26 | 94 | | 15 | 105 | | 22 | 98 | | 18 | 102 | |
| Adjuvant chemotherapy | Yes | 8 | 55 | 0.55 | 4 | 59 | 0.59 | 7 | 56 | 0.73 | 4 | 59 | 0.42 |
| | No | 26 | 138 | | 14 | 150 | | 21 | 143 | | 16 | 148 | |
| Molecular subtype (basal vs. luminal) | luminal | 41 | 169 | **0.029** | 25 | 185 | **0.01** | 35 | 175 | **0.048** | 27 | 183 | **0.04** |
| | basal | 14 | 118 | | 5 | 127 | | 12 | 120 | | 8 | 124 | |

Comparison of clinical parameters between AMP and NONAMP patients for DNA Amplicon Core, DNA Amplicon Ext1, DNA Amplicon Ext2, and RNA Amplicon Core. Significant p-values are highlighted in **bold.**

**Table 3. Univariable and multivariable analysis of Overall Survival (OS) and Disease-Free Survival (DFS) for amplicon definitions.**

| Clinicopathologic variables | | Univariable | | | | Multivariable | | | |
|---|---|---|---|---|---|---|---|---|---|
| | | Overall survival | | Disease-Free Survival | | Overall survival | | Disease-Free Survival | |
| | | HR (95% CI) | P-value | HR (95% CI) | P-value | HR (95% CI) | P-value | HR (95% CI) | P-value |
| Age | ≥70 vs <70 | **1.63 [1.19; 2.23]** | **0.002** | 1.23 [0.86; 1.77] | 0.25 | **1.76 [1.24; 2.49]** | **0.002** | | |
| Gender | Male vs Female | 0.81 [0.58; 1.14] | 0.23 | 0.92 [0.62; 1.36] | 0.66 | | | | |
| T stage | T3/4 vs T≤2 | **1.99 [1.36; 2.92]** | **<0.001** | **2.59 [1.67; 4.03]** | **<0.001** | 1.5 [0.98; 2.32] | 0.065 | **2.31 [1.41; 3.79]** | **<0.001** |
| Lymph node status | N+ vs N0 | **2.11 [1.52; 2.93]** | **<0.001** | **2.38 [1.64; 3.46]** | **<0.001** | **2.05 [1.44; 2.91]** | **<0.001** | **2 [1.36; 2.92]** | **<0.001** |
| Molecular subtype | (basal vs. luminal) | 1.28 [0.92; 1.77] | 0.14 | 1.24 [0.85; 1.8] | 0.26 | 1.19 [0.85; 1.7] | 0.31 | | |
| DNA_Amplicon_Core | AMP vs NONAMP | 0.91 [0.59; 1.4] | 0.66 | 1.1 [0.7; 1.76] | 0.66 | | | | |
| RNA_Amplicon_Core | AMP vs NONAMP | 0.6 [0.32; 1.14] | 0.12 | 1.04 [0.59; 1.82] | 0.88 | 0.64 [0.32; 1.32] | 0.22 | | |
| DNA_Amplicon_Ext1 | AMP vs NONAMP | 0.79 [0.49; 1.28] | 0.34 | 1.09 [0.67; 1.78] | 0.73 | | | | |
| DNA_Amplicon_Ext2 | AMP vs NONAMP | 0.74 [0.42; 1.31] | 0.3 | 1.05 [0.6; 1.84] | 0.86 | | | | |
| RNA_AMP_COX6C | HIGH vs LOW | 0.66 [0.43; 1] | **0.04** | 0.83 [0.53; 1.29] | 0.4 | 0.76 [0.47; 1.22] | 0.25 | | |
| RNA_AMP_OSR2 | HIGH vs LOW | 1.54 [0.98; 2.43] | 0.06 | **1.97 [1.2; 3.22]** | **0.01** | 1.41 [0.86; 2.32] | 0.17 | **1.91 [1.15; 3.16]** | **0.01** |

Univariable and multivariable analysis of Overall survival (OS) and Disease-Free Survival (DFS), including clinicopathologic characteristics, amplicon definitions, and gene definitions. Variables from univariable analysis with a p-value of <0.2 were entered into a multivariable model. P-values with a statistical significance of <0.05 are highlighted in **bold**.

## Validation of the biomarkers COX6C and OSR2

The median gene expression of normalized Ct values of OSR2 and COX6C was significantly different between both genes (COX6C: 30.62; OSR2: 34,53; p<0.001; S7 Fig). OSR2 mRNA overexpression (≥ median) showed a trend towards lower DFS rates (HR [CI] 4.54 [0.93; 22.16]; p = 0.06) and predicted significantly worse OS rates (HR [CI] 6.25 [1.37; 28.38]; p = 0.018). In contrast to improved survival in the in silico analysis, COX6C overexpression (≥ median) showed a trend towards worse OS (HR [CI] 2.6 [0.92; 7.29]; p = 0.07) and DFS (HR [CI] 3.05 [0.81; 11.52]; p = 0.099) in RT-qPCR data (Fig 4). Interestingly, selecting the 3rd quartile as a cut-off point for COX6C overexpression resulted in a significant decrease in OS (HR [CI] 4.49 [1.69; 11.91]; p = 0.0026) and DFS (HR [CI] 4.29 [1.28; 14.42]; p = 0.018). In multivariable analyses COX6C overexpression (≥ median) significantly predicted OS (HR [CI] 3.41 [1.05; 11.1];

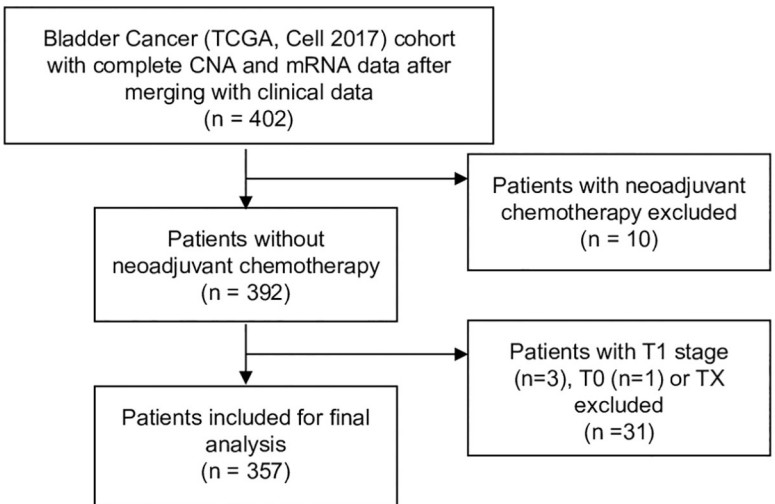

**Fig 1. REMARK diagram of the study cohort based on the TCGA cell 2017 MIBC cohort.**

**Univariable analysis of DFS of individual genes according to RNA_Amplicon_Core**

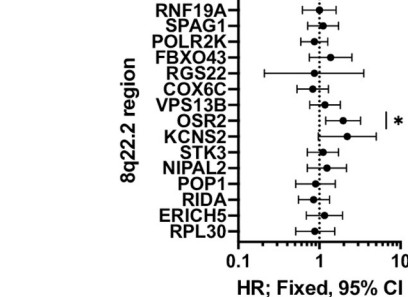

**A**

**Univariable analysis of OS of individual genes according to RNA_Amplicon_Core**

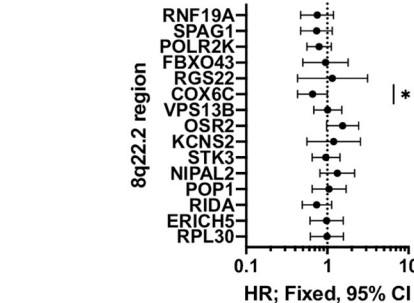

**B**

**Fig 2. Univariable analysis of OS and DFS of individual genes according to RNA_AMP_Gene.** (A) Forest plot depicting hazard ratios (HR) with 95% confidence intervals (95% CI) of DFS of RNA_HIGH_Gene versus RNA_LOW_Gene. Genes of the 8q22.2 region are arranged according to their genomic location on 8q22.2. * indicates a p-value of 0.01. (B) Forest plot depicting hazard ratios (HR) with 95% confidence intervals (95% CI) of OS of RNA_HIGH_Gene versus RNA_LOW_Gene patients. Genes of the 8q22.2 region are arranged according to their genomic location on 8q22.2. * indicates a p-value of 0.04.

p = 0.04). In contrast to the in silico data, OSR2 overexpression ($\geq$ median) was not an independent prognostic factor for DFS (HR [CI] 4.3 [0.8; 23.21]; p = 0.09) (S11–S15 Tables).

Kaplan-Meier regressions showing (A) OS of OSR2 mRNA expression $\geq$ median vs. < median; (B) OS of COX6C mRNA expression $\geq$ median vs. < median; (C) DFS of OSR2 mRNA expression $\geq$ median vs. < median and (D) DFS of COX6C mRNA expression $\geq$ median vs. < median.

Note that an mRNA expression greater than the median is associated with a significantly lower OS for OSR2 (p = 0.0066) and COX6C (p = 0.0287). OSR2 $\geq$ median is also associated with a significantly reduced DFS (p = 0.0399).

## Association of copy number alterations and mRNA levels in the 8q22.2 region

Concordance of DNA and mRNA level could only be assessed between the two core amplicon definitions, and was significant between both definitions (p<0.001). Twenty-three patients were identified as AMP by both 8q22.2 amplicon core definitions, with RNA_Amplicon_Core showing a higher sensitivity (23 of 30 RNA_AMP_Core patients vs. 23 out of 58 CNA_AMP_-Core patients). DNA_Amplicon_Core was associated with higher specificity (292 CNA_NO-NAMP_Core out of 299 patients vs. RNA_NONAMP_Core 292 out of 327).

A classical amplicon definition would suggest an increase in transcript levels in DNA amplified samples. As anticipated, *RNF19A* and *SPAG1* showed a higher median gene expression in the

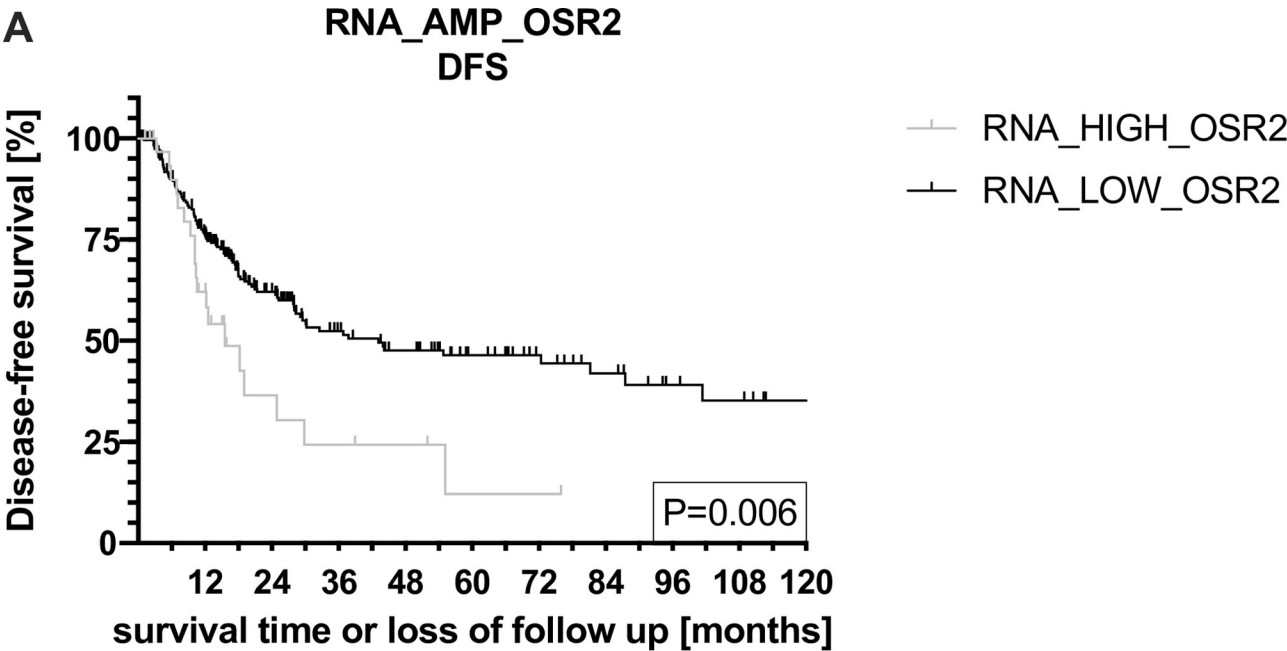

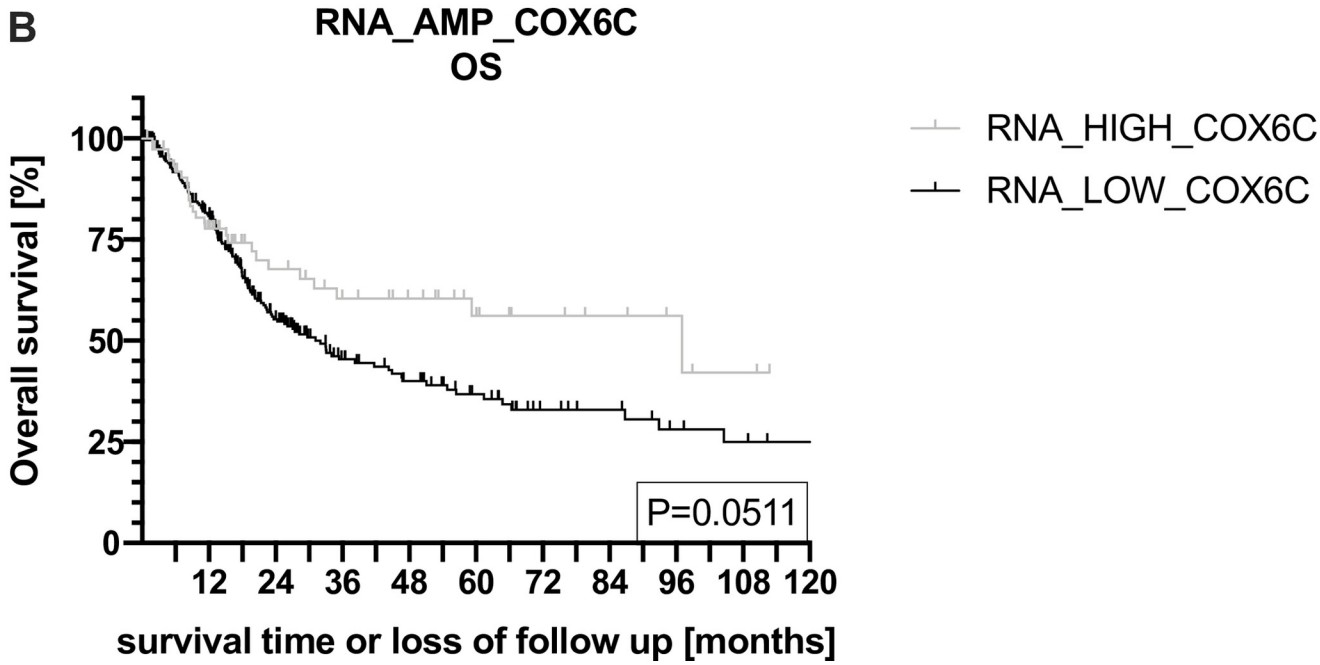

**Fig 3. COX6C and OSR2 are independent prognostic factors for survival in silico.** (A) Kaplan-Meier regression showing disease-free survival (DFS) of RNA_HIGH_OSR2 vs. RNA_LOW_OSR2. RNA_HIGH_OSR2 is associated with a statistically significant worse DFS compared to NONAMP (p = 0.006). (B) Kaplan-Meier regression showing overall survival (OS) of RNA_HIGH_COX6C vs. RNA_LOW_COX6C. RNA_HIGH_COX6C is associated with improved OS compared to RNA_LOW_COX6C (p = 0.0511).

AMP group. A marked difference between both groups could also be observed for *POLR2K* (ENSG00000147669), *COX6C*, and *RIDA* (ENSG00000132541). Interestingly, *RGS22* (ENSG 00000132554), *KCNS2* (ENSG00000156486), and *NIPAL2* (ENSG00000104361) showed negative gene expression in both groups. Several genes also showed a switch from negative gene expression in the NONAMP group to positive gene expression in the AMP group (Fig 5, S3 Table) [38].

Median z-scores of mRNA gene expression of DNA_Amplicon_Core are shown in gray for AMP and black for NONAMP. Genes are arranged in descending order based on the percentage of copy number amplifications (GISTIC2 ≥ 2) in MIBC. P-values: **<0.001, *0.018 for OSR2 and NS (p-value for RGS22 = 0.89; KCNS2 = 0.11 and NIPAL2 = 0.77.

Spearman coefficient analysis of correlations of mRNA expression of genes revealed a strong correlation within the core region (Rho = 0.64, p<0.001). *POLR2K*, *COX6C*, and *RIDA2* were highly correlated (COX6C - POLR2K, Rho = 0.63 p<0.001; RIDA–COX6C, Rho = 0.54 p<0.001; RIDA–POLR2K, Rho = 0.64; p<0.001). Interestingly, *NIPAL2* and *VPS13B* showed the highest correlation overall (Rho = 0.69; p<0.001). *COX6C* and *ERICH5* showed multiple weak negative correlations with several genes (S10 Table).

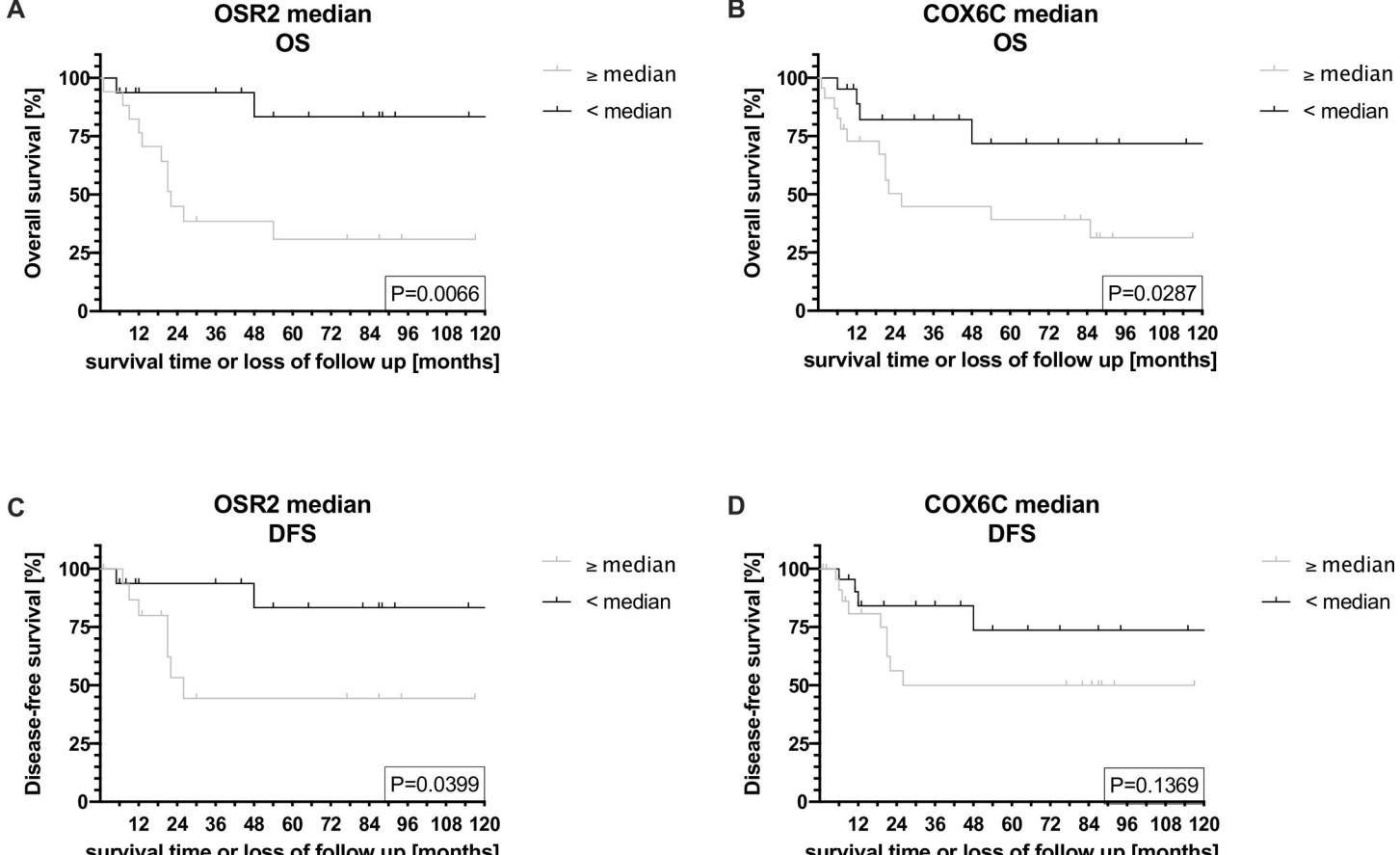

**Fig 4. COX6C and OSR2 are independent prognostic factors for survival in RT-qPCR data.**

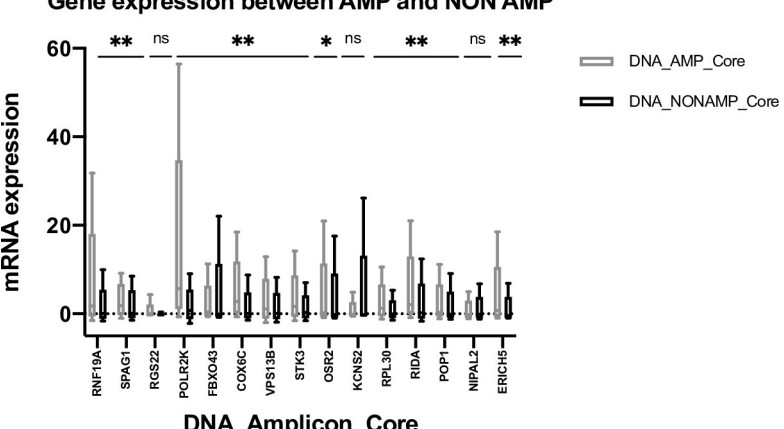

**Fig 5. Median mRNA expression of DNA_Amplicon_Core between AMP and NONAMP groups.**

## Associations with genomic events and gene functions of the 8q22.2 amplicon

When analyzing mutations in 18 known oncogenes and tumor suppressor genes, the DNA_AMP_Core showed a higher rate of mutations in TP53 (65% vs. 46%) and KMT2A (19% vs. 10%), as compared to the DNA_NONAMP_Core. In contrast, FGFR3, CREBBP and ERBB2 were more frequently mutated in the DNA_NONAMP_Core group (DNA_AMP_Core vs. DNA_NONAMP_Core; FGFR3: 4% vs. 15%; CREBBP: 4% vs. 13%; ERBB2: 4% vs. 13%). (S5 Table)

Interestingly, there was a significant difference in the mutational status of ERBB2 between the RNA_AMP_Core and RNA_NONAMP_Core. No patient in the RNA_AMP_Core group showed mutations in ERBB2, whereas 13% of RNA_NONAMP_Core patients exhibited mutations in this gene (p = 0.04). (S6 and S7 Tables)

Amplification of 8q22.2 was significantly associated with gains in 6p22.3, 1q23.3, 8q22.3, 3p25.2, 19q12, 1q21.2 and 7p21.1. Associated deletions included 1p36.11 and 3p14.2. 1q23.3, 8q22.3, 3p25.2 and 19q12 remained significant after correction for multiple testing was performed (S8 Table). Biological and functional gene analysis revealed most genes in the 8q22.2 region to be involved in protein or RNA/DNA processing pathways (S5 Table).

*RNF19A* and *FBXO43* (ENSG00000156509) are both involved in protein ubiquitination, whereas *POLR2K* mediates the transcription of DNA into RNA. *COX6C* and *STK3* (ENSG00000104375) show potential involvement in apoptosis, as cytochrome oxidases can initiate the apoptotic process; however, *STK3* produces a pro-apoptotic kinase, which is part of the Hippo signaling pathway.

## Discussion

### Comprehensive molecular analysis within multiple levels of the genetic code

This is the first in-depth comprehensive molecular characterization of the 8q22.2 region in the MIBC genome using the TCGA cell 2017 cohort. This study highlights the importance of placing comprehensive characterization at the center of any analysis regarding the potential role of restricted genomic regions in cancers. In this study, a comprehensive approach was used to take the structure and genomic proximity of the region into account, and it included multiple levels of the genetic code to allow for a more accurate assessment of interactions and gene

activity. Once a restricted genomic region has been structurally defined, its correlation with phenotype and prognostic impact on survival can be assessed using clinicopathologic and survival data. A similar concept has already been used by Jones et al., who used integrative analysis of gene copy number alterations and mRNA expression to identify potential drivers of tumor recurrence in breast cancer [39].

## Genes of the 8q22.2 region exhibit typical and paradoxical patterns of copy number alterations and RNA expression levels

While data from breast cancer and other cancers suggest that amplified CNA leads to subsequent mRNA overexpression, our results show that genes of the 8q22.2 region express a heterogenous concordance pattern between CNA and mRNA gene expression levels [38, 40–42]. While the classical pattern of CNA amplification leading to mRNA overexpression seems to be conserved in the core region of RNF19A and SPAG1, genes such as KCNS2 and NIPAL2 demonstrate a contrasting pattern, with a higher median gene expression in non-amplified patients. This paradoxical pattern of opposing CNA and mRNA expression levels has been described in lung adenocarcinoma by Tokar et al., who proposed that epigenetic regulatory mechanisms were responsible for these changes [43]. These mechanisms include microRNA-mediated control of mRNA expression levels and methylation.

Genomic proximity of genes does not seem to be responsible for the heterogenous concordance pattern observed between CNA and mRNA in 8q22.2, as KCNS2 and NIPAL2 exhibit a paradoxical pattern and lie in the middle of the region, with STK3 between them, exhibiting the classical CNA-mRNA expression pattern.

Additionally, potential tumor growth and progression in the CNA_AMP_Core group could have increased through *TP53* and KMT2A mutations, as this group harbored a higher percentage of these mutations than did the CNA_NONAMP_Core. When investigating head and neck squamous cell cancer cell lines, Cheng et al. found concurrent 3q26.3 amplification and *TP53* mutation to be associated with a reduced survival rate [44]. Additionally, Zhang et al. found that KMT2A promoted melanoma cell growth by targeting the hTERT signalling pathway [45]. Interestingly, FGFR3, CREBBP and ERBB2 were more frequently mutated in the NONAMP group. Future research into the associations of 8q22.2 amplifications and mutations in bladder cancer is needed.

Amplification of 8q22.2 could also be viewed in an overarching context of chromosomal aberrations, as amplified patients showed significantly more frequent gains of 1q23.3, 8q22.3, 3p25.2 and 19q12, even after correcting for multiple testing. Concomitant gain of 3p25.2 was also reported by Hurst et al. [21]. Findings by Riester et al. further support the association of amplifications in specific regions of MIBC with survival, since gains in the 1q23.3 region were associated with poor survival in two cohorts of metastatic urothelial carcinoma [15].

Additional investigations are necessary to determine whether 8q22.2 amplification is a relevant genomic event in the pathogenesis of BC or simply a bystander of more complex chromosomal changes.

## RNA_HIGH_OSR2 is an independent prognostic factor for survival

Analysis of the prognostic impact of individual genes of the 8q22.2 region revealed OSR2 to be an independent prognostic factor for survival. RNA_HIGH_OSR2 was associated with a worse prognosis for DFS (HR [CI]; 0.5 [0.32; 0.85]; p = 0.01) in the in silico analysis and decreased OS (HR [CI]; 6.25 [1.37; 28.38]; p = 0.018) in RT-qPCR data.

While the exact role of *OSR2* in tumorigenesis is unknown, its methylation appears to play a role. Li et al. investigated the diagnostic value of *OSR2* hypermethylation in gastric cancer

patients and found significant differences in methylation status between cancer samples and normal controls. The authors concluded that hypermethylation of *OSR2* and two other genes could provide a good alternative for non-invasive detection of gastric cancer [46]. Kostareli et al. found that promoter hypermethylation of *ALDH1A2*, *OSR2*, *GATA4*, *GRIA4*, and *IRX4* showed a significant inverse correlation to transcript levels in oropharyngeal squamous cell carcinoma (OPSCC). A signature of low methylation levels in *ALDH1A2* and *OSR2* promoters, as well as high methylation levels in *GATA4*, *GRIA4*, and *IRX4* promoters correlated well with improved survival in 3 independent patient cohorts [47].

Thus, the promotor hypermethylation status of OSR2 could be a potential predictor of survival in MIBC and a potential target of epigenetic therapies. In addition to copy number amplifications on the 8q22.2 OSR2 promotor, methylation could potentially influence OSR2 mRNA expression. Further research is needed to investigate this association.

## Amplicon definitions are associated with unique respective clinicopathologic variables

Comparative analysis should include two gene expression levels (CN and mRNA) in correlations with phenotype and prognostic impact on survival. This is important because the prognostic value of the 8q22.2 core region, containing amplifications of RNF19A and SPAG1, seems to be highly dependent on the chosen amplicon definition. A potential relationship between an mRNA defined amplicon and lymph node status may exist, as RNA_AMP_Core patients were more likely to show lymph node positive disease. This is consistent with research conducted by Lindquist et al., who found gains in 22 genes on chr3p25 and chr11p11, which remained significantly associated with lymph node involvement in MIBC. The addition of CNA data improved discrimination relative to the use of clinical variables alone (p = 0.04). Gains in chr3p25 and chr11p11 were further associated with shorter overall survival periods [14].

Associations of chemotherapy resistance and 8q22 amplification have been shown by other groups. Clearly additional experiments are needed to address the potential involvement of 8q22.2 in treatment resistance [48, 49].

Furthermore, luminal subtype MIBC may be associated with the presence of amplicons, as patients with core and extended amplicons in the 8q22.2 region were more likely to harbor a luminal subtype. The luminal infiltrated subtype shows a mesenchymal expression signature [7]. Thus, genes on 8q22.2 involved in mesenchymal differentiation, such as *COX6C*, could play a role in the pathogenesis of luminal MIBC. Further studies in this regard are essential in order to understand the underlying mechanisms.

This study has several weaknesses, including a lack of uniformity in techniques, algorithms, bioinformatics, and definitions used for the in silico investigation of amplicons in cancer.

Results of this study should be interpreted cautiously due to the heterogeneity of the cohorts, the small sample size of the validation cohort and the use of varying overexpression definitions (z-scores and median). Validation in larger independent cohorts is needed.

Additionally, we could not address the relevance of balanced or unbalanced gains because loss of heterozygosity was not assessed.

Nonetheless, our study is the first one to highlight the significance of the 8q22.2 region in MIBC and to identify potential targets in this region.

## Conclusions

High OSR2 mRNA levels are associated with decreased DFS in silico and OS in RT-qPCR data, respectively, and could serve as a potential biomarker for MIBC. Although

RNA_Amplicon_Core mRNA expression is high in lymph node positive patients, future research on this association is necessary.

The typical pattern of CNA and high mRNA expression could not be validated in every gene of the 8q22.2 region, as some genes showed an inverse relationship between CNA and mRNA expression.

The associations found in this study were detected through an in-depth and comprehensive molecular characterization, which included multiple levels of the genetic code and correlations with phenotype and prognostic impact on survival.

Clearly, this approach needs to be further validated in large independent cohorts and other genomic regions of MIBC.

## Supporting information

**S1 Fig. Oncoprint of 357 patients from the TCGA Cell 2017 MIBC cohort.** Red bars show copy number amplifications according to GISTIC2.0. Blue bars represent deep deletions. RN7SL350P, RN7SKP85 and SNORA72 were excluded from further analysis due to a lack of mRNA gene expression data. The graphic was created using cBioPortal's oncoprint tool. (PDF)

**S2 Fig. Oncoprint of 357 patients from the TCGA Cell 2017 MIBC cohort.** Red frames represent mRNA overexpression (z-score $\geq 2$) and blue frames show low mRNA expression. RN7SL350P, RN7SKP85 and SNORA72 were excluded from further analysis due to a lack of mRNA gene expression data. The graphic was created using cBioPortal's oncoprint tool. (PDF)

**S3 Fig. Overall survival and disease-free survival of the GISTIC2.0 and the mRNAz2 core regions between AMP and NONAMP.** (A) Kaplan-Meier regression showing overall survival (OS) of patients with GISTIC2 core amplicons (AMP) vs. patients without GISTIC2 core amplicons (NONAMP). (B) Kaplan-Meier regression showing overall-survival (OS) of patients with mRNAz2 core amplicons (AMP) vs. patients without mRNAz2 core amplicons (NON-AMP). (C) Kaplan-Meier regression showing disease-free survival (DFS) of patients with GIS-TIC2 core amplicons (AMP) vs. patients without GISTIC2 core amplicons (NONAMP). (D) Kaplan-Meier regression showing disease-free survival (DFS) of patients with mRNAz2 core amplicons (AMP) vs. patients without mRNAz2 core amplicons (NONAMP). AMP and NONAMP are shown in black and grey, respectively. No statistical differences between groups were observed. (5-year OS: AMP vs. NONAMP, p-value; DNA_Amplicon_Core: 39% vs. 42%; p = 0.66; RNA_Amplicon_Core: 51% vs. 40%; p = 0.12; DNA_Amplicon_Ext1: 46% vs 40%; p = 0.91; DNA_Amplicon_Ext2: 49% vs. 40%; p = 0.3)(5-Year DFS: AMP vs. NONAMP, p-value; DNA_Amplicon_Core: 36% vs. 45%; p = 0.66; RNA_Amplicon_Core: 35% vs. 45%; p = 0.89; DNA_Amplicon_Ext1: 41% vs 43%; p = 0.73; DNA_Amplicon_Ext2: 46% vs. 43%; p = 0.86). (PDF)

**S4 Fig. Overall survival and disease-free survival of the GISTIC2.0 extended 1 and GIS-TIC2.0 extended 2 between AMP and NONAMP.** (A) Kaplan-Meier regression showing disease-free survival (DFS) of patients with GISTIC2 extended 1 amplicons (AMP) vs. patients without GISTIC2 extended 1 amplicons (NONAMP). (B) Kaplan-Meier regression showing disease-free survival (DFS) of patients with GISTIC2 extended 1 amplicons (AMP) vs. patients without GISTIC2 extended 1 amplicons (NONAMP). (C) Kaplan-Meier regression showing overall survival (OS) of patients with GISTIC2 extended 1 amplicons (AMP) vs. patients without GISTIC2 core amplicons (NONAMP). (D) Kaplan-Meier regression showing overall

survival (OS) of patients with GISTIC2 extended 2 amplicons (AMP) vs. patients without GIS-TIC2 extended 2 amplicons (NONAMP). AMP and NONAMP are shown in black and grey, respectively. No statistical differences between groups were observed.
(PDF)

**S5 Fig. Distribution of lymph node stage in relation to varying degrees of somatic copy number changes and gene expression levels.** Distribution of lymph node status (%) is stratified according to 4 combinations of copy number amplification or deletion and high (z-score $\geq$ 2) or low (z-score $<$ 2) mRNA expression. Lymph node positive and negative stage are almost evenly distributed for the combination of amplification and overexpression, while all other combinations show a greater proportion of negative lymph node stage.
(PDF)

**S6 Fig. Coamplification of amplified genes varies based on chosen amplicon definition.** Genes of the 8q22.2 region are arranged according to their genomic location and distributed accross amplicon definitions based on the extent of the percentage of coamplification. DNA_Amplicon_Ext1 and DNA_Amplicon_Ext2 extend beyond the core region of RNF19A and SPAG1, to include seven genes from RNF19A to VPS13B (DNA_Amplicon_Ext1) and 15 genes from RNF19A to ERICH5 (DNA_Amplicon_Ext2).
(PDF)

**S7 Fig. Normalized gene expression of COX6C and OSR2.** Distribution of normalized 40-$\Delta$Ct values for COX6C and OSR2. Median gene expression was compared using the Mann-Whitney test and was significantly different between both genes (COX6C: 30.62; OSR2: 34,53; p<0.001).
(PDF)

**S1 Document. TCGA Cell 2017 in silico cohort data.**
(XLSX)

**S2 Document. RT-qPCR data and clinical data from the Clinic for Urology and Urosurgery at the University Hospital Mannheim.**
(XLSX)

**S1 Table. Univariable analysis of mRNA amplicon definitions with different z-score cut-offs.**
(DOCX)

**S2 Table. Multivariable analysis of Overall survival of mRNA amplicon with z-score cut-off z = 1.**
(DOCX)

**S3 Table. Median gene expression of the core CNA amplicon in AMP and NAMP.**
(DOCX)

**S4 Table. Subgroup analysis of molecular subtypes in the TCGA cohort.**
(DOCX)

**S5 Table. Functions of genes in the 8q22.2 region.**
(DOCX)

**S6 Table. Rate of mutations in AMP and NONAMP patients for DNA_Amplicon_Core.**
(DOCX)

**S7 Table. Rate of mutations in AMP and NONAMP patients for RNA_Amplicon_Core.**
(DOCX)

**S8 Table. Association of gains at 8q22.2 with other genomic events.**
(DOCX)

**S9 Table. Sequence of primers and probes used in the RT-qPCR.**
(DOCX)

**S10 Table. Spearman coefficient analysis of genes in the 8q22.2 region.**
(DOCX)

**S11 Table. Univariable OS and DFS analysis of COX6C and OSR2 in vitro in a cohort from the university hospital Mannheim (n = 46).**
(DOCX)

**S12 Table. Univariable analysis of clinicopathologic features of the cohort from the university hospital Mannheim (n = 46 patients).**
(DOCX)

**S13 Table. Multivariable analysis of OS for COX6C in the university hospital Mannheim cohort (n = 46).**
(DOCX)

**S14 Table. Univariable analysis of DFS for OSR2 in the university hospital Mannheim cohort (n = 33).**
(DOCX)

**S15 Table. Multivariable analysis of OS and DFS for OSR2 in the university hospital Mannheim cohort.**
(DOCX)

## Acknowledgments

We would like to thank Annette Steidler for excellent technical assistance and edit911 (https://edit911.com) for professional language editing. We would like to thank the colleagues from the Institute of Pathology and the Clinic of Urology and Urosurgery of the University Hospital Mannheim for providing the tumor samples and great assistance with the research.

## Author Contributions

**Conceptualization:** Daniel Uysal, Philipp Erben.

**Data curation:** Daniel Uysal, Zoran V. Popovic.

**Investigation:** Daniel Uysal.

**Methodology:** Daniel Uysal.

**Project administration:** Philipp Erben.

**Resources:** Zoran V. Popovic.

**Software:** Daniel Uysal.

**Supervision:** Philipp Erben.

**Writing – original draft:** Daniel Uysal, Philipp Erben.

**Writing – review & editing:** Karl-Friedrich Kowalewski, Maximilian Christian Kriegmair, Ralph Wirtz, Zoran V. Popovic, Philipp Erben.

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
