## [Decision Letter · Decision Letter 0]

3 Sep 2020

PONE-D-20-18560

Comprehensive Molecular Characterization of the 8q22.2 region identifies prognostic relevance of COX6C and OSR2 mRNA in muscle invasive bladder cancer

PLOS ONE

Dear Dr. Erben,

Thank you for submitting your manuscript to PLOS ONE. After careful consideration, we feel that it has merit but does not fully meet PLOS ONE’s publication criteria as it currently stands. Therefore, we invite you to submit a revised version of the manuscript that addresses the points raised during the review process. Most importantly, in addition to replying to all specific comments, you should report whether the findings are replicated in independent datasets, thus providing robustness to your data.

Please submit your revised manuscript within 6 months. If you will need more time than this to complete your revisions, please reply to this message or contact the journal office at plosone@plos.org. Please include the following items when submitting your revised manuscript:

We look forward to receiving your revised manuscript.

Kind regards,

Francisco X. Real

Academic Editor

PLOS ONE

Journal Requirements:

2. In the Methods section, we ask that you please provide any accession numbers of the datasets downloaded from TCGA and cBioPortal for your study.

We note that one or more of the authors are employed by a commercial company: STRATIFYER Molecular Pathology GmbH.

3.1. Please provide an amended Funding Statement declaring this commercial affiliation, as well as a statement regarding the Role of Funders in your study. If the funding organization did not play a role in the study design, data collection and analysis, decision to publish, or preparation of the manuscript and only provided financial support in the form of authors' salaries and/or research materials, please review your statements relating to the author contributions, and ensure you have specifically and accurately indicated the role(s) that these authors had in your study. You can update author roles in the Author Contributions section of the online submission form.

3.2. Please also provide an updated Competing Interests Statement declaring this commercial affiliation along with any other relevant declarations relating to employment, consultancy, patents, products in development, or marketed products, etc.  

6. Please ensure that you refer to Figure 4 in your text as, if accepted, production will need this reference to link the reader to the figure.

Reviewers' comments:

Reviewer's Responses to Questions

**Comments to the Author**

1. Is the manuscript technically sound, and do the data support the conclusions?

Reviewer #1: Partly

Reviewer #2: No

2. Has the statistical analysis been performed appropriately and rigorously? 

Reviewer #1: I Don't Know

Reviewer #2: No

3. Have the authors made all data underlying the findings in their manuscript fully available?

Reviewer #1: Yes

Reviewer #2: Yes

4. Is the manuscript presented in an intelligible fashion and written in standard English?

Reviewer #1: Yes

Reviewer #2: No

5. Review Comments to the Author

Reviewer #1: In this manuscript the authors evaluated the possible role of amplification in 8q22.2 region in bladder cancer. Using TCGA data they found that this region shows CNV in around 10% of patients uin parallel with RNA amplification of the corresponding genes. among them, they can identify COX6C and OSR2 mRNA in correlation with specific clinicopthological characteristics.

In general the manuscript is well written and the conclusions are supported by the data. Nonetheless a possible concern comes from the fact that only TCGA data are used in the analyses. The clinicla annotation of this dataset can be controversial, therefore a possible validation including other cohorts is strictly recommended.

In particular the possible validation of the identified genes by RTqPCR in other cohorts is highly relevant.

Another aspect of potential controversial is the association with specific molecular subtypes. According the manuscript the authors observed a possible association with luminal tumors. However in TCGA there are many other subtypes besides luminal and basal BC. Therefore, it is difficult to ascertain how the aithors have performed this analysis. A more detailed study including specific markers is recommended.

Minor aspects,

The use of a z-score normalization value ≥2 is appropriate but a supplementary table providing values using more or less restrictive z-scores would help to realize the relevance of the amplicon

Reviewer #2: Uysal et al performed in silico analysis of data from 361 MIBC patients from TCGA to investigate the prognostic role of 8q22.2. The authors investigate amplicons using DNA and RNA based methods, and finally analyze genes at 8q22.2 that may be of clinical relevance.

1. Introduction: The authors should decide if this is a paper describing novel biological/clinical features associated the 8q22.2 region in bladder cancer – or if this is a more technical paper, describing the use of the methods used. This could be rewritten to focus on one or the other. It is not clear when reading the introduction what the main scope is. The final part of the introduction (line 99) should be rewritten.

2. The main focus of the paper- chromosomal region 8q22.2 – is mentioned in the introduction as amplified in the cbioportal.org website. However, as little is known about this in BC I suggest to move this to the result section. Overall, 8q22.2 may be interesting, but it is not clear why the authors choose to investigate this. This should be clearly written and expanded upon in the manuscript.

3. Methods: it is not clear if CNV data is derived from SNP arrays or from exome sequencing. Please specify. Also, as CNV is one of the main points in the manuscript, the authors should describe the different types og gains (high vs low gain, and imbalanced vs balanced). Also – how does the level of amplification affect the level of expression?

4. Are the gains at 8q22.2 associated with other genomic events? This may indicate that this event is an artefact, especially if gains are balanced.

5. Methods: patients receiving NAC are excluded. The reason for this is not clear – the full cohort receive various treatments, and delayed chemotherapy upon metastatic disease, and adjuvant chemotherapy. Please comment on this.

6. Page 6 – there seem to be redundancy in the text regarding the patient population exclusion criteria.

7. Methods: how was the z-score of >=2 chosen for mRNA overexpression ? Clarify the definition of “RNA_HIGH_GeneX” and “RNA_LOW_GeneX”.

8. Methods + Table 2: Why include variables with a p-value < 0.2 in the multivariable analysis?

9. Line 184: The authors call a p-value < 0.2 significant. However, they also state in line 192 that p-values < 0.05 were considered significant. Please clarify.

10. Line 138: the reason for splitting the 8q22.2 into a “core “ and “extended” region is not clear. An explanation is provided in line 146-147, but does this represent a continuous part of DNA at 8q22.2, or simply representation of genes where no correlation between DNA and RNA is observed throughout the region? Furthermore, the results related to this is presented in the method section. (Table 1).

11. Table 2: Luminal vs basal subtype is a strong dichotomization of the data, and it is not strange that this becomes significant. But it would be more interesting to see how the regions actually correlated to the consensus classification for MIBC.

12. Table 2: According to the supplementary excel file, no tumors below T2 are included. However, tumors are divided into T-stage ≤2 and T stage 3/4 in Table 2 and in line 153,201,229. The ≤2 T stage indicates that non-muscle invasive tumors are included and is therefore misleading. Rephrase this to T2 and not ≤T2.

13. N-stage is only significant for RNA and not for DNA, which indicated that the expression is not driven by amplification. In my opinion, it would be stronger to make a joint analysis of genes in the region – affected by amplification, and simultaneously overexpressed. As I read the paper this is not the case, and this is investigated separately. The authors touch upon this in the section starting line 278, but it is not clear why this is not performed in general through the manuscript – if the purpose is to identify genes of interest affected by gene amplification and with elevated expression.

14. Correlation to mutations and gene functions: the authors correlate to 18 gene mutations. Was correction for multiple testing performed here ? The significance reported is not high, so it would be important to correct for this.

15. Finally, the authors should investigate other public datasets, in order to validate the findings – e.g. only RNA data to validate specific genes.

6. PLOS authors have the option to publish the peer review history of their article (what does this mean?). If published, this will include your full peer review and any attached files.

Reviewer #1: No

Reviewer #2: No

---

## [Author Response · Author response to Decision Letter 0]

11 Jan 2021

Please find our responses to specific reviewer and editor comments in the attached file: response to reviewers.

---

## [Decision Letter · Decision Letter 1]

8 Feb 2021

PONE-D-20-18560R1

A comprehensive molecular characterization of the 8q22.2 region reveals the prognostic relevance of OSR2 mRNA in muscle invasive bladder cancer

PLOS ONE

Dear Dr. Erben,

Thank you for submitting your manuscript to PLOS ONE. After careful consideration, we feel that it has merit but does not fully meet PLOS ONE’s publication criteria as it currently stands. Therefore, we invite you to submit a revised version of the manuscript that addresses the points raised during the review process. The comments are minor and the paper might be acceptable without further external review.

Please submit your revised manuscript within one month. If you will need more time than this to complete your revisions, please reply to this message or contact the journal office at plosone@plos.org. Please include the following items when submitting your revised manuscript:

We look forward to receiving your revised manuscript.

Kind regards,

Francisco X. Real

Academic Editor

PLOS ONE

Reviewers' comments:

Reviewer's Responses to Questions

**Comments to the Author**

1. If the authors have adequately addressed your comments raised in a previous round of review and you feel that this manuscript is now acceptable for publication, you may indicate that here to bypass the “Comments to the Author” section, enter your conflict of interest statement in the “Confidential to Editor” section, and submit your "Accept" recommendation.

Reviewer #1: All comments have been addressed

Reviewer #2: All comments have been addressed

2. Is the manuscript technically sound, and do the data support the conclusions?

Reviewer #1: Yes

Reviewer #2: Yes

3. Has the statistical analysis been performed appropriately and rigorously? 

Reviewer #1: Yes

Reviewer #2: Yes

4. Have the authors made all data underlying the findings in their manuscript fully available?

Reviewer #1: Yes

Reviewer #2: Yes

5. Is the manuscript presented in an intelligible fashion and written in standard English?

Reviewer #1: Yes

Reviewer #2: Yes

6. Review Comments to the Author

Reviewer #1: (No Response)

Reviewer #2: Authors have addressed my previous concerns adequately.

However, I have a few comments that may improve the manuscript further:

1. line 126: I think the use of "in silico" i headline and in line 127 is a bit strange. I would rewrite and just indicate that data has been produced earlier.

2. line 152: rewrite to just mention a validation cohort - it is not important to highlight the origin in the headline.

3. line 284: remove the "cell 2017" from the headline - and "the" before TCGA as it is already included in the acronym..

4. line 325: remove "in silico" again

5. line 382: remove "in silico"

7. PLOS authors have the option to publish the peer review history of their article (what does this mean?). If published, this will include your full peer review and any attached files.

Reviewer #1: **Yes: **Jesús M Paramio

Reviewer #2: No

---

## [Author Response · Author response to Decision Letter 1]

10 Feb 2021

Response to Reviewers 

1. line 126: I think the use of "in silico" i headline and in line 127 is a bit strange. I would rewrite and just indicate that data has been produced earlier

We thank Reviewer #2 for his suggestion. The headline and line 127 have been modified accordingly. 

Text passage: 

“TCGA cohort 

Data for the first muscle invasive bladder cancer (MIBC) cohort for this study was derived from The Cancer Genome Atlas and has been produced in earlier analyses [7].”

2. line 152: rewrite to just mention a validation cohort - it is not important to highlight the origin in the headline

The headline has been modified.

Text passage:

“Validation cohort”

3. line 284: remove the "cell 2017" from the headline - and "the" before TCGA as it is already included in the acronym..

The headline has been modified according to the suggestions or Reviewer #2.

Text passage:

“8q22.2 is a highly amplified region in TCGA cohort and can be divided into a core and extended region “

4. line 325: remove "in silico" again & 5. line 382: remove "in silico"

We thank Reviewer #2 for his suggestions. “In silico” has been omitted in the text passages.

Text passage: 

“Univariable and multivariable survival analysis (DFS and OS) of the 8q22.2 region and genes in the TCGA cohort”

“Validation of the biomarkers COX6C and OSR2”

---

## [Editor Report · Decision Letter 2]

25 Feb 2021

A comprehensive molecular characterization of the 8q22.2 region reveals the prognostic relevance of OSR2 mRNA in muscle invasive bladder cancer

PONE-D-20-18560R2

Dear Dr. Erben,

We’re pleased to inform you that your manuscript has been judged scientifically suitable for publication and will be formally accepted for publication once it meets all outstanding technical requirements.

Kind regards,

Francisco X. Real

Academic Editor

PLOS ONE
---

## [Editor Report · Acceptance letter]

3 Mar 2021

PONE-D-20-18560R2 

A comprehensive molecular characterization of the 8q22.2 region reveals the prognostic relevance of OSR2 mRNA in muscle invasive bladder cancer 

Dear Dr. Erben:

I'm pleased to inform you that your manuscript has been deemed suitable for publication in PLOS ONE. Congratulations! Your manuscript is now with our production department. 

Kind regards, 

on behalf of

Dr. Francisco X. Real 

Academic Editor

PLOS ONE